# Analysis of the Physical Characteristics of an Anhydrous Vehicle for Compounded Pediatric Oral Liquids

**DOI:** 10.3390/pharmaceutics15112642

**Published:** 2023-11-20

**Authors:** Daniel Banov, Yi Liu, Kendice Ip, Ashley Shan, Christine Vu, Oleksandr Zdoryk, August S. Bassani, Maria Carvalho

**Affiliations:** 1Professional Compounding Centers of America (PCCA), Houston 77099, TX, USA; 2Institute for Pharma Technology, School of Life Sciences, University of Applied Sciences und Arts Northwestern Switzerland, 4132 Muttenz, Switzerland; 3Department of Pharmaceutical Technologies and Medicines Quality Assurance, Institute of the Professional Skills Improvement in the Field of Pharmacy, National University of Pharmacy, 61002 Kharkiv, Ukraine

**Keywords:** pediatric drug delivery, extemporaneous preparations, innovative technologies, pharmaceutical development, anhydrous vehicle, sedimentation stability, thixotropic behavior, droplet size, self-emulsifying properties, content uniformity

## Abstract

The paucity of suitable drug formulations for pediatric patients generates a need for customized, compounded medications. This research study was set out to comprehensively analyze the physical properties of the new, proprietary anhydrous oral vehicle SuspendIt^®^ Anhydrous, which was designed for compounding pediatric oral liquids. A wide range of tests was used, including sedimentation volume, viscosity, droplet size after dispersion in simulated gastric fluid, microscopic examination and content uniformity measurements to evaluate the properties of the anhydrous vehicle. The results showed that the vehicle exhibited consistent physical properties under varying conditions and maintained stability over time. This can be attributed to the unique blend of excipients in its formulation, which not only maintain its viscosity but also confer thixotropic behavior. The unique combination of viscous, thixotropic and self-emulsifying properties allows for rapid redispersibility, sedimentation stability, accurate dosing, potential drug solubility, dispersion and promotion of enhanced gastrointestinal distribution and absorption. Furthermore, the vehicle demonstrated long-term sedimentation stability and content uniformity for a list of 13 anhydrous suspensions. These results suggest that the anhydrous oral vehicle could serve as a versatile base for pediatric formulation, potentially filling an important gap in pediatric drug delivery. Future studies can further investigate its compatibility, stability and performance with other drugs and in different clinical scenarios.

## 1. Introduction

Pharmacists around the world continue to prepare extemporaneously a wide range of compounded medications as the availability of commercial medications for some groups of adults and children remains inadequate despite ongoing efforts to improve [1,2,3,4,5,6]. Hospitals often require age-appropriate dosage forms with safe excipients formulated with one or more active pharmaceutical ingredients (APIs) at various strengths [7,8,9,10]. Amongst the most used dosage forms for pediatric patients are oral liquids, particularly suspensions [11,12,13,14,15,16]. Another important consideration for oral liquid medications is their potential suitability for feeding tube administration [17,18]. Today, there is a lack of commercially available liquid medications specifically formulated for tube-feeding purposes [19,20]. Therefore, hospital pharmacists considering compounding drugs for feeding tube administration must be well informed about the various factors involved in preparing such medications. These factors include the formulation compatibility with the materials used in the feeding tube, the stability of the drug in the liquid formulation (pH, temperature and exposure to light), the size of the drug particles, the compatibility of the medication with the nutritional formula or vehicle, the administration technique (tube type and size), the taste and palatability of the medication and its safety and sterility [20,21,22,23,24]. Examples of the most common drug groups and corresponding APIs that are prepared in pharmacies for different age groups include the following: gastroenterological (calcium carbonate, hyoscine butyl bromide, hyoscine hydrobromide, loperamide hydrochloride, omeprazole), cardiovascular (captopril, nifedipine, phenoxybenzamine hydrochloride), antibacterial/antiprotozoal (metronidazole, doxycycline, isoniazid, neomycin,), diuretic (hydrochlorothiazide and spironolactone), anticonvulsant (clonazepam, gabapentin, phenobarbital), analgesic and anti-inflammatory drugs (diclofenac, morphine, codeine, naproxen, paracetamol) [12,13,14,15].

Oral vehicles play a critical role in the formulation of oral liquid medicines and are particularly important in addressing the needs of special patient populations [25]. These vehicles are used to create solutions and suspensions of water-soluble and water-insoluble drugs, providing the desired stability, viscosity, pH and taste-masking capabilities. In addition, oral vehicles are crucial for advancing the development of new molecules, and could facilitate the creation of specialized, non-clinical oral formulations that can handle unique characteristics in the research and development phase. They can also act as solubilizing excipients in oral formulations, helping to improve drug solubility and bioavailability [26].

There is a limited supply of anhydrous oral vehicles for pharmaceutical compounding [27], and the selection of anhydrous vehicles is rather limited, with only a few options, such as Unispend Anhydrous^®^, SuspendRx™ Anhydrous and the newly developed anhydrous oral vehicle SuspendIt^®^ Anhydrous available. Unispend Anhydrous is a plant-based, anhydrous oral suspending vehicle that is naturally sweetened. It contains medium-chain triglycerides, glyceryl distearate and polyglyceryl-3 oleate as its key ingredients. The vehicle also includes a bitterness-masking agent to enhance its palatability. This vehicle is especially suitable for APIs that are unstable in water, are lipophilic or have unknown aqueous stability [25,28]. SuspendRx Anhydrous shares many characteristics with Unispend Anhydrous, including its plant-based, anhydrous formulation, stability profile and intended applications. SuspendRx Anhydrous does not differ significantly in its specific ingredient composition, which includes medium-chain triglycerides, lipids, glycerides, bitter-block technology and stevia for added sweetness [29]. SuspendIt Anhydrous vehicle is designed to be self-emulsifying when in contact with gastric fluids, to exhibit thixotropic behavior for easy redispersion and to offer improved palatability. As an oily vehicle, it incorporates a carefully selected mix of components, including medium-chain triglycerides, a wetting agent, emulsifiers, stabilizers, antioxidants and sweeteners. From a nutritional standpoint, SuspendIt Anhydrous alone contains 0.3% protein, 87.7% fat and 7.7% carbohydrates [30]. 

The development of alternative vehicles is important for several reasons. First, anhydrous vehicles offer advantages as depot or reservoir vehicles for lipophilic or poorly water-soluble APIs [25]. Such vehicles can be used for slow-release drug patterns over an extended period of time, which is beneficial for patients requiring long treatments [31]. Anhydrous vehicles provide greater flexibility in formulating various drug compounds due to their compatibility with lipophilic compounds, a wide range of options for excipients, potential enhancement of permeation and facilitation of the formulation of sensitive compounds. This flexibility allows pharmacists to tailor the formulation to specific patient needs, ensuring accurate dosing and optimal therapeutic outcomes. This is particularly important for pediatric patients, who routinely require age-appropriate dosage forms formulated with appropriate vehicles. In addition, anhydrous vehicles can improve the stability of compounded preparations, ensuring that the drugs remain chemically, physically and microbiologically stable over time [32]. The absence of water in the dosage form minimizes the risk of microbial growth, which also plays a key role in dosage form instability and reduction in potency and overall quality. Anhydrous formulations can be designed to be compatible with the materials of nasogastric (feeding) tubes. This characteristic ensures that patients who require enteral nutrition can receive their medications via a feeding tube without compromising the drug’s stability or therapeutic efficacy. In addition, anhydrous vehicles can improve the palatability of oral drugs by providing taste-masking capabilities [33]. This taste enhancement can be achieved via various mechanisms, such as controlling the solubility of the API, altering the texture and mouthfeel, creating a mouth-coating effect, modulating interactions with taste receptors as compared to aqueous solutions, avoiding hydrolysis and ensuring homogeneous distribution and a balanced taste profile of the lipophilic compounds. The palatability of oral liquid medications is crucial for patient compliance, particularly in children, and understanding the taste preferences and considerations of patients can help healthcare professionals make informed decisions in prescribing and counselling. Therefore, it is crucial to conduct an evaluation of the taste characteristics of the vehicles for preparing compounded oral liquids [34,35]. In the dynamic field of pharmaceutical compounding, innovations are infrequent but pivotal. 

In order to achieve the necessary properties of oral liquids, anhydrous vehicles need to be characterized by a certain set of physical characteristics, which must be investigated during the development of the vehicle. The physical properties have a significant impact on quality, stability, organoleptic properties and use (Appendix A) [25,36,37]. Liquid vehicles can be formulated to provide the desired osmolarities, viscosities, pH and taste-masking capabilities, which are important factors in matching the intentions of the pharmacist [35]. Sedimentation analysis allows us to assess the settling behavior of the vehicle, which can impact the even distribution of drug particles within the formulation [38,39,40]. The evaluation of thixotropic properties helps to understand the viscosity changes in the vehicle during agitation, enabling the rapid redispersion of APIs with minimal sedimentation [41,42]. The assessment of droplet size in the context of self-emulsifying drug delivery systems (SEDDS) plays a pivotal role, particularly when these anhydrous suspensions interact with an aqueous phase, aided by gentle mixing similar to gastric fluid and gastrointestinal (GI) movements, to form finely dispersed emulsions. The fine and uniform droplets of formed emulsions play an important role in enhancing drug absorption and bioavailability in the intestinal environment. Droplet size evaluation provides insights into the APIs distribution, bioavailability, stability, appearance and texture of the formulation and the rate of release [43,44]. The emulsifying and dispersing properties of vehicles are essential for improving the solubility, dispersibility and absorption of hydrophobic drugs [43,45,46]. Self-emulsifying drug delivery systems create a spontaneous emulsion when they come into contact with water. Microscopic evaluation and fluorescence microscopy techniques allow the self-emulsifying properties of the vehicle to be investigated. Ensuring content uniformity is critical for accurate dosing and consistent drug delivery, which are of paramount importance in feeding tube administration and pediatric compounding [47].

The objective of this study is to conduct a thorough investigation into the physical properties of this newly developed anhydrous oral vehicle by evaluating its sedimentation behavior, thixotropic properties, droplet size distribution after dispersion in simulated gastric fluid, self-emulsifying properties and content uniformity. The aim of this study is to gain a comprehensive understanding of the vehicle’s performance and potential applications in compounding oral liquids for pediatrics.

## 2. Materials and Methods

The SuspendIt Anhydrous (will be referred to as the Test Vehicle in subsequent sections of this article) and SuspendIt (will be referred to as the Water-Based Test Vehicle in subsequent sections of this article) vehicles, along with the APIs used in this comprehensive analysis, were provided by the Professional Compounding Centers of America (PCCA). For comparative purposes, two proprietary anhydrous vehicles were used, each incorporating a triglyceride emulsification system. The above vehicles were used to evaluate and compare their physical properties with the Test Vehicle (further in the text, Comparator 1 and Comparator 2), providing valuable insight into its performance as a potential oral vehicle for compounded pediatric oral liquids.

The following materials were used: gabapentin USP (#30-4213 PCCA, Houston, TX, USA); enrofloxacin USP (#30-4815 PCCA, Houston, TX, USA); Acesulfame potassium FCC (#30-4398 PCCA, Houston, TX, USA); steviol glycosides 95% (#30-4539 PCCA, Houston, TX, USA); sodium chloride USP granular (#30-4280 PCCA, Houston, TX, USA); flavor, banana cream artificial (#30-2169 PCCA, Houston, TX, USA); flavor, raspberry artificial (#30-2323 PCCA, Houston, TX, USA); chloroquine phosphate tablets, USP (Rising Pharmaceuticals, Inc. Saddle Brook, NJ, USA); doxycycline hyclate USP (30-3129, PCCA); ketotifen Fumarate EP (30-4345, PCCA); metronidazole USP (30-1449, PCCA); nifedipine USP (55-5205, PCCA); phenoxybenzamine hydrochloride USP (55-3119, PCCA); tretinoin USP (30-1270, PCCA), simulated gastric fluid (RICCA Chemical Company #7108-2.5, Arlington, TX ); fluorescein sodium (#F6377, Sigma-Aldrich, St. Louis, MO, USA); curcumin powder 95% (#30-3497, PCCA, Houston, TX, USA).

The selection of formulas and APIs is based on the most frequently downloaded and used formulas from the PCCA database, which specifically relate to suspensions formulated with non-aqueous bases. This focus is highly relevant to compounded paediatric oral liquids and ensures that the study is directly applicable to current pharmacy compounding practices. Suspensions were prepared for each formulation according to the established compounding guidelines. A measured amount of an API was accurately weighed and placed in a mortar. The powder was then carefully mixed with a small amount of the chosen vehicle to form a uniform paste. Additional vehicle was gradually added to the mortar and the resulting mixture was transferred into a 500 mL volumetric flask using a rubber spatula. This transfer process was repeated three times to ensure complete transfer of the liquid from the mortar. The volumetric flask was filled to the calibration mark with the chosen vehicle. The mixture was vortexed using a vortex mixer and subjected to sonication in an ultrasonic bath for 5 min to remove air bubbles. After removing any air bubbles, the volumetric flask was again filled to the calibration mark with vehicle. Finally, the flask was placed on a magnetic stirrer for further homogenization.

### 2.1. Sedimentation Behavior Investigation by Visual Analysis

The formulations (gabapentin 100 mg/mL anhydrous suspension and enrofloxacin 100 mg/mL anhydrous suspension) were selected based on several key factors: their solubility in water, particle size and the concentration of the active ingredient in the formulation. The least favourable conditions for sedimentation stability were chosen to subject the anhydrous oral vehicles to the most rigorous testing. Gabapentin (water soluble) was included due to its specific particle size distribution with D10 at 27.3 µm, D50 at 11.6 µm and D90 at 283.9 µm. This selection allows us to investigate how well the anhydrous vehicle can accommodate water-soluble APIs with a wide range of particle sizes. In contrast, enrofloxacin (water-insoluble) with a D90 particle size of 88.7 µm provides an opportunity to evaluate the effectiveness of the anhydrous oral vehicle in suspending water-insoluble compounds.

Three 100 mL portions of gabapentin 100 mg/mL anhydrous suspension and enrofloxacin 100 mg/mL anhydrous suspension were prepared using the Test and Comparator Vehicles. The unpleasant taste of the APIs was masked with the flavoring agents banana cream (yellow color) for gabapentin and raspberry (pink color) for enrofloxacin, and with sweetener (steviol glycosides). The suspensions were placed in a standing position at room temperature 20 ± 2 °C in the dark place for 29 days. On day 28, the glass containers were shaken vigorously with the same amount of force and duration, and allowed to settle again for 24 h. The results were recorded using a camera Canon EOS 5D Mark IV (Canon Inc., Tokyo, Japan). The sedimentation volume was determined in triplicate at each sampling point using the following equation:
F = V_u_/V_0_,(1)
where V_u_ represents the final volume of sediment as the suspension settles, and V_0_ represents the original volume of the suspension [37].

### 2.2. Viscosity and Thixotropic Properties

The viscometer used was a Brookfield Viscometer DV-II + Pro (Stresstech, CANNON Instrument Company, State College, PA, USA). For the evaluation of the viscosity of the Test Vehicle and Comparators 1 and 2, a bob and cup CC25 assembly was employed, with a shear rate of 0.01 1/S at 25 °C for each of the vehicle products. 

The thixotropic properties of the Test Vehicle were evaluated and compared to the industry-standard Water-Based Test Vehicle, which contains water. The Water-Based Test Vehicle serves as a water content oral vehicle designed to offer a unique blend of functionalities, including thixotropic behavior and a patented synergistic polymer complex. The composition and ingredients of this vehicle are detailed in a study by Ip et al. (2018) [48]. From a formulation standpoint, the vehicle is strategically designed to offer desirable characteristics such as ease of redispersion, taste enhancement and long shelf life, making it suitable for a wide range of applications. 

The thixotropic properties were determined using the viscometer equipped with an RV-2 spindle. Both vehicles were carefully mixed in a 600 mL beaker for 5 min at a speed of 700 rpm and then allowed to stand for 10 min. Each sample was measured every 10 s, the shear rate applied ranged from 0.1 to 100 1/s and the temperature of each of the samples was 20 ± 2 °C. The reported viscosity values represent the average of 12 measurements.

### 2.3. Content Uniformity

There were 13 anhydrous suspensions, each containing a single API dispersed in the Test Vehicle, selected for the assessment of content uniformity: chloroquine phosphate 100 mg/5 mL, doxycycline 50 mg/mL, doxycycline 100 mg/mL, enrofloxacin 10 mg/mL, enrofloxacin 100 mg/mL, ketotifen 1 mg/mL, metronidazole 50 mg/mL, nifedipine 4 mg/mL, phenoxybenzamine hydrochloride 2 mg/mL and tretinoin 10 mg/mL. These suspensions were prepared using the classical technique described above. The tretinoin 10 mg/mL and doxycycline 100 mg/mL also were prepared using the FlackTek SpeedMixer DAC 1200-500 (FlackTek, Inc., Landrum, SC, USA) for comparative analysis.

The evaluation of content uniformity was performed in line with the specifications of USP general chapter 〈905〉, UNIFORMITY OF DOSAGE UNITS. The acceptance value (AV) for the content uniformity was determined using the following formula:
AV = |M − X_m_| + ks,(2)
where M—reference value, X_m_—mean of the individual contents, k—acceptability constant, s—standard deviation of the sample.

Potency testing was conducted using an Ultra-Performance Liquid Chromatography (UPLC) assay as defined by USP chapter 〈621〉, CHROMATOGRAPHY. All samples were analyzed in a Waters Acquity H-Class Bio UPLC System, consisting of a Separations Module, Column Manager, Heater/Cooler, Autosampler and Waters Acquity UPLC PDA Detector (Waters Inc., Charlotte, NC, USA). Volumetric aliquots for quantification were drawn from the middle of the containers without touching the inner surface. The detailed chromatographic conditions for the analysis of the content uniformity of anhydrous suspensions with different APIs and gradient profiles are given in Appendix A.

### 2.4. Droplet Size

Laser diffraction was used to study the size of the droplets in the emulsions obtained after shaking the test suspensions with water. Droplet size analysis was performed using a Saturn DigiSizer II 5205 (Micromeritics, Norcross, GA, USA). Testing was performed in accordance with ISO 13320 standards [49]. The study involved the preparation of compounded anhydrous oral suspensions containing ibuprofen at a concentration of 40 mg/mL using both the Test Vehicle and Comparator 2 and comparing the results with a commercial ibuprofen (water-based) [50] reference product at the same concentration. The test samples were vortexed for approximately 30 s. Subsequently, 10 g of each suspension was dispersed in 15 mL of simulated gastric fluid, followed by 1 h of stirring.

### 2.5. Microscopic Evaluation of the Self-Emulsifying Properties

The self-emulsifying properties of the Test Vehicle were evaluated in two stages: microscopic evaluation and fluorescence microscopy. In both tests, the Test Vehicle was compared to Comparators 1 and 2. For the microscopic evaluation, metronidazole 50 mg/mL anhydrous suspensions were prepared using the Test Vehicle and Comparators 1 and 2. The three suspensions were mixed in water at a 1:1 ratio (*v*/*v*). Afterward, a sample of each suspension was observed under a light microscope with a total magnification of 100× for evaluation of the droplet size formation. 

Fluorescent microscopy was used for the evaluation of the distribution pattern of the hydrophilic substance fluorescein sodium (high-solubility), and also the lipophilic substance curcumin (low-solubility and high-permeability), in comparison to the performance of the comparison vehicles. The fluorescein sodium 1% anhydrous suspensions and curcumin 1% anhydrous suspensions were prepared including the Test Vehicle, as well as Comparators 1 and 2. All suspensions were gently mixed with water at a 1:1 ratio (*v*/*v*). A fluorescence microscopy test was performed on all suspensions using a fluorescent light and, when applicable, a white light with 4× and 10× objective lenses. A Nikon Eclipse TS100 inverted phase microscope equipped with the NIS-Elements imaging software version 5.02 (Nikon, Tokyo, Japan) and a Lumencor^®^ MIRA Light Engine (4-NII-FA) were used for the study. A filter set with excitation and emission spectra of 460–490 nm and 500–560 nm was used for fluorescence excitation.

### 2.6. Palatability Evaluation

A taste and impression survey was conducted to evaluate the palatability of the Test Vehicle. A total of 68 adult participants completed the survey during a taste panel session of the PCCA International Seminar 2022, a seminar held in Houston, Texas, for compounding pharmacists from the US and abroad. Participants were asked to rate the taste, mouthfeel, sweetness, odor, color and overall impression of the anhydrous vehicle on a 5-point scale from “excellent” to “very poor” via Google Forms. Additionally, comments from the participants were collected and taken into consideration to gain further insights into their experiences with the anhydrous vehicle. The data obtained from the survey were then analyzed using Microsoft Excel (Version 2205, Microsoft, USA).

## 3. Results

### 3.1. Sedimentation Behaviour Investigation Using Visual Analysis

At day 0, all suspensions were at 100% volume, indicating their initial homogeneity. However, as the study progressed, sedimentation was observed in all formulations. The photos of the suspensions throughout the 29-day period are presented in Figure 1. By day 7, the suspensions prepared with Comparator 1 showed a significant decrease in volume, with only 55% and 52% remaining for gabapentin and enrofloxacin, respectively. In comparison, the Test Vehicle and Comparator 2 showed better resistance to sedimentation, retaining 92% and 80% for gabapentin and 94% and 81% for enrofloxacin, respectively. 

At day 28, the sedimentation volumes continued to decrease for all suspensions, indicating progressive settling of the suspended particles. The most significant sedimentation was observed in the formulations prepared with Comparator 1, with only 42% remaining for gabapentin and 46% for enrofloxacin. In contrast, the Test Vehicle and Comparator 2 exhibited higher stability, retaining 88% and 67% for gabapentin and 86% and 74% for enrofloxacin, respectively. However, after shaking on day 28, all suspensions resuspended completely. After 24 h of redispersion (day 29), an improvement in sedimentation stability was observed for all formulations. Test Vehicle and Comparator 2 showed complete recovery, reaching 100% volume for both gabapentin and enrofloxacin. Comparator 1, although showing improvement, remained slightly lower at 95% for gabapentin and 99% for enrofloxacin. 

### 3.2. Viscosity and Thixotropic Properties

Viscosity measurements showed the following results: Water-Based Test Vehicle—2.33 Pa·s, Test Vehicle—86.57 Pa·s, Comparator 1—17.41 Pa·s, Comparator 2—22.42 Pa·s. The viscosity of the Test Vehicle was observed to be 37 times higher than that of the Water-Based Test Vehicle. Compared to Comparators 1 and 2, the Test Vehicle had a viscosity four to five times higher only. This substantial difference in viscosity can have a significant effect on the sedimentation stability, flow behaviour and overall performance of these vehicles. 

In line with the results of previous studies by Pramar Y. [51] and Graves R. [52] investigating the thixotropic nature of the Test Vehicle, our experiment aimed to compare the thixotropic indices by evaluating the viscosity at low to high shear rates for both aqueous and anhydrous vehicles. The results showed that the Test Vehicle was 20–40 times more viscous and exhibited similar synergistic thixotropic properties to the Water-Based Test Vehicle, as shown in Figure 2 (left versus right). When a low shear rate (0.1 1/s) was applied for the first 200 s, both vehicles maintained their resting viscosity (blue lines). However, when the shear rate was increased (100 1/s), the viscosity decreased dramatically to almost 0 cP (red lines). Subsequently, when the shear rate returned to a low level (0.1 1/s), both vehicles rapidly regained their original resting viscosity within seconds.

### 3.3. Content Uniformity

There were 130 suspension samples analyzed for eight different APIs. The mean potency of the samples of suspensions varied from 95.50 to 104.60%. The mean result for all samples was 100.83% with a standard deviation of 1.18%. Potency testing showed that none of the anhydrous suspensions deviated from the 90.0–110.0% potency specification (Table 1). A comparative analysis of the two preparation methods, the conventional mortar-and-pestle technique and the modern FlackTek SpeedMixer approach, was carried out on two suspensions, doxycycline 100 mg/mL and tretinoin 10 mg/mL. For the doxycycline 100 mg/mL suspension, the mean potency was found to be 100.00% using the mortar-and-pestle method and 102.00% using the FlackTek SpeedMixer. The RSD was 1.50% and 1.89% and the acceptance value was 5.10% and 5.03%, respectively. Similarly, for the tretinoin 10 mg/mL suspension, the mean potency was 100.74% using the mortar and pestle method and 102.00% using the FlackTek SpeedMixer. The RSD was 0.88% and 1.20% and the acceptance value was 4.34% and 3.37%, respectively.

### 3.4. Droplet Size

The droplet size diameter values are listed for three percentile points: 90%, 50% and 10%. The 90% value indicates that 90% of the droplet diameters are below the listed value. Similarly, the 50% and 10% values indicate the diameters below which 50% and 10% of the drops lie, respectively. The Mean ± Std Dev column shows the mean droplet size together with its standard deviation (Table 2). The analysis showed that the majority of the ibuprofen droplets in the Test Vehicle had diameters ranging from approximately 48.5 to 65.4 μm. These results closely paralleled the droplet size distribution observed in the commercial reference product for ibuprofen 40 mg/mL. In stark contrast, the droplets in the Comparator 2 formulation had significantly larger diameters, ranging from approximately 192.7 to 255.3 μm.

### 3.5. Microscopic Evaluation of the Self-Emulsifying Properties

When the anhydrous metronidazole 50 mg/mL suspensions were mixed with water, the Test Vehicle suspension produced a uniform emulsion, indicating its potential to enhance drug absorption. In contrast, suspensions using the Comparators were found to be immiscible with water, resulting in the formation of two distinct phases (Appendix A). When the samples were examined under a microscope at 4× magnification, the suspension using the Test Vehicle again showed a homogeneous dispersion (Figure 3). The suspension from Comparator 1 showed aggregation of metronidazole and the suspension from Comparator 2 showed separation of the water and oil phases (Figure 3B,C).

Fluorescein sodium and curcumin were selected to determine the influence of the nature of the API (hydrophilic and lipophilic) on the drug distribution after ingestion of anhydrous suspension or after mixing with aqueous beverages. Hydrophilic and lipophilic APIs are routinely used in clinical practice, making it critical to understand the properties of the vehicle. The results showed that when the anhydrous suspension was mixed with water at a 1:1 (*v*/*v*) ratio, the Test Vehicle exhibited unique self-emulsifying properties by forming an emulsion. This consisted of dispersed lipophilic droplets in an aqueous continuous phase. Fluorescein sodium was solubilized in the aqueous phase and everything came together to form an emulsion (Figure 4). Conversely, no homogeneous emulsion was formed with the suspensions of Comparators 1 and 2 as most of the fluorescein sodium remained soluble in water with a noticeable separation between the water and oil phases (Figure 4B,C).

As expected, the water-insoluble curcumin was entrapped within the oil droplets, resulting in the formation of a uniform emulsion with the Test Vehicle (Figure 5). For Comparators 1 and 2, the curcumin was mainly dispersed in the oil phase, and the separation between the oil and water phases was observed (Figure 5B,C).

### 3.6. Palatability Evaluation

More than 80% of the participants rated the taste, sweetness, mouthfeel, color and odor positively (excellent and good). Seven (10.29%) respondents were dissatisfied with the base odor. Some comments also provided valuable insights into specific preferences and considerations, highlighting the advantages and flavor particularities of the vehicle, such as no aftertaste, no slimy mouthfeel, slight throat irritation and sweet taste. These observations confirm the focus of the vehicle design, as the taste has been specifically formulated to be more palatable to pediatric patients. Overall, the sensory characteristics of the base were rated quite positively by almost all respondents, with 63 (92.65%) rating them as excellent and good. The results of the palatability analysis were presented in the form of a cumulative chart, which depicted the distribution of ratings for each characteristic (Figure 6).

## 4. Discussion

The results of the sedimentation behavior study showed that both the gabapentin and enrofloxacin suspensions prepared using the Test Vehicle showed good suspension stability over the 29-day period. The homogeneity observed during the study indicated the effectiveness of the anhydrous vehicle in preventing particle settling. Despite some sedimentation, the suspensions were easily resuspended by shaking them, indicating their ability to maintain the desired pharmaceutical quality. The absence of separation after redispersion on day 29 further confirmed the stability and homogeneity of the suspensions. Comparator 2 showed significantly better commendable performance in resisting sedimentation in relation to Comparator 1. Gabapentin displayed stronger sedimentation in the Comparator 1 and 2 vehicles than enrofloxacin, which correlates with the larger particle size of the former API. The inclusion of medium-chain triglycerides and lipids in the composition of the three different vehicles appears to contribute to their improved sedimentation stability. However, due to the unique composition and excipients of the Test Vehicle, it exhibits greater sedimentation stability. The primary factors influencing sedimentation rates in anhydrous vehicles include API particle size, property differences between the particles and the liquid medium, vehicle viscosity, temperature and drug–excipient interactions. After controlling for variables such as API particle size, temperature and the nature of the base and API, the most distinguishing factor appears to be the significantly higher viscosity of the Test Vehicle. This increased viscosity is primarily due to its unique excipient composition, making it a focus for potential applications in compounded pediatric oral liquids. These results suggest that the target anhydrous Test Vehicle is suitable for oral liquid formulation and provides stable suspensions with minimal sedimentation.

The observed thixotropic properties of both vehicles indicate the ability of their suspensions to thin on shaking and thicken on standing. The thinning behavior on shaking is of particular importance as it allows for rapid redispersion of the APIs, ensuring precise dose uniformity and facilitating accurate measurement and administration of each dose. In addition, thickening on standing plays a critical role in minimizing API sedimentation once the suspensions settle, thus ensuring the superior physical stability of the preparations. The presence of thixotropic properties is also indicated by the results of the sedimentation volume analysis of the gabapentin and enrofloxacin suspensions. The study validates the thixotropic nature of the Test Vehicle via a comprehensive evaluation of its viscosity and shear rate behavior, establishing its comparability with the Water-Based Test Vehicle. These inherent thixotropic properties hold substantial promise in promoting the homogeneous dispersion of APIs, ensuring precise dosing and enhancing the long-term stability of the formulated suspensions.

All samples of the compounded suspensions showed minimal variation in content and all AVs did not exceed the L1 criteria of 15.0%, indicating that all suspensions met the USP requirements for content uniformity. Despite the fact that the FlackTek SpeedMixer method showed a slightly higher mean potency for both suspensions, the content uniformity of the suspensions prepared using both methods meets the criteria. Although the variance in potency between the two methods of preparation is not significant and within acceptable limits, the FlackTek SpeedMixer may offer an advantage in terms of speed and ease of preparation.

The Test Vehicle appeared to facilitate the production of smaller droplet sizes after dispersion in gastric fluid for ibuprofen, similar to the commercial reference product and Comparators. This result suggests the potential for more efficient distribution in the GI tract and improved absorption of the compounded anhydrous oral suspensions.

The SEDDS of the Test Vehicle allow for the formation of a uniform emulsion, characterized by small and homogeneous droplets. As a result, it is expected that the self-emulsifying properties of the Test Vehicle are likely to promote increased drug solubility, dispersibility, absorption and bioavailability. Conversely, Comparators 1 and 2 did not show any self-emulsifying properties. Microscopic observations clearly showed the presence of metronidazole crystals in the sample of Comparator 1 and distinct water and oil droplets in the sample Comparator 2, indicating suboptimal solubility and dispersibility, and, by inference, potentially poorer bioavailability. 

The results of the taste and impression survey showed that the majority of participants rated the taste, mouthfeel, sweetness and odor of the anhydrous vehicle positively. The cumulative graph visually showed the distribution of ratings, indicating that a significant proportion of participants found the characteristics of the anhydrous vehicle to be excellent or good. However, there were also a few participants who mentioned certain limitations, such as the aftertaste or a specific odor. The positive feedback from participants and the overall impression of the anhydrous vehicle suggests its potential as a suitable base for compounded oral liquids.

## 5. Conclusions

This study highlights the importance of comprehensive knowledge for the correct selection of the appropriate vehicle when compounding oral liquids for pediatrics. When developing new anhydrous vehicles for the extemporaneous preparation of medications, it is important to study sedimentation volume, viscosity, droplet size after dispersion in gastric fluid and microscopic evaluation, as these tests significantly demonstrate drug dispersion and absorption, which are critical to achieving the desired quality, safety and efficacy of the corresponding compounded medications. 

The study showed that the viscosity and thixotropic properties of the Test Vehicle played a critical role in its performance, facilitating easy redispersion while minimizing sedimentation, confirming the success of the excipient selection. The ability of the vehicle to thin on shaking and thicken on standing is essential to ensure dose uniformity and ease of administration, which are particularly critical factors in pediatric formulations.

The use of the FlackTek SpeedMixer, although not significantly affecting the content uniformity, can offer advantages in terms of speed and ease of preparation. The unique self-emulsifying properties of the Test Vehicle, distinguished from those of the Comparators, suggest an enhanced potential for improved drug solubility and dispersibility as a consequence of absorption. This is particularly promising for the delivery of poorly water-soluble APIs, contributing to their therapeutic efficacy.

The Test Vehicle stands out as a robust, effective and patient-friendly base for the formulation of compounded pediatric oral liquids. Its unique composition, characterized by increased viscosity and excellent thixotropic properties, positions it as a strong candidate for improving the stability, dosing accuracy and overall acceptability of anhydrous suspensions. The current study has shown that it not only meets but in many aspects exceeds the requirements for such a vehicle, making it a focus for future applications in this specialized area of pharmacy. However, further investigations regarding its compatibility with various APIs, stability studies, in vitro nasogastric feeding tube testing and its palatability to the pediatric population are recommended.

## Figures and Tables

**Figure 1 pharmaceutics-15-02642-f001:**
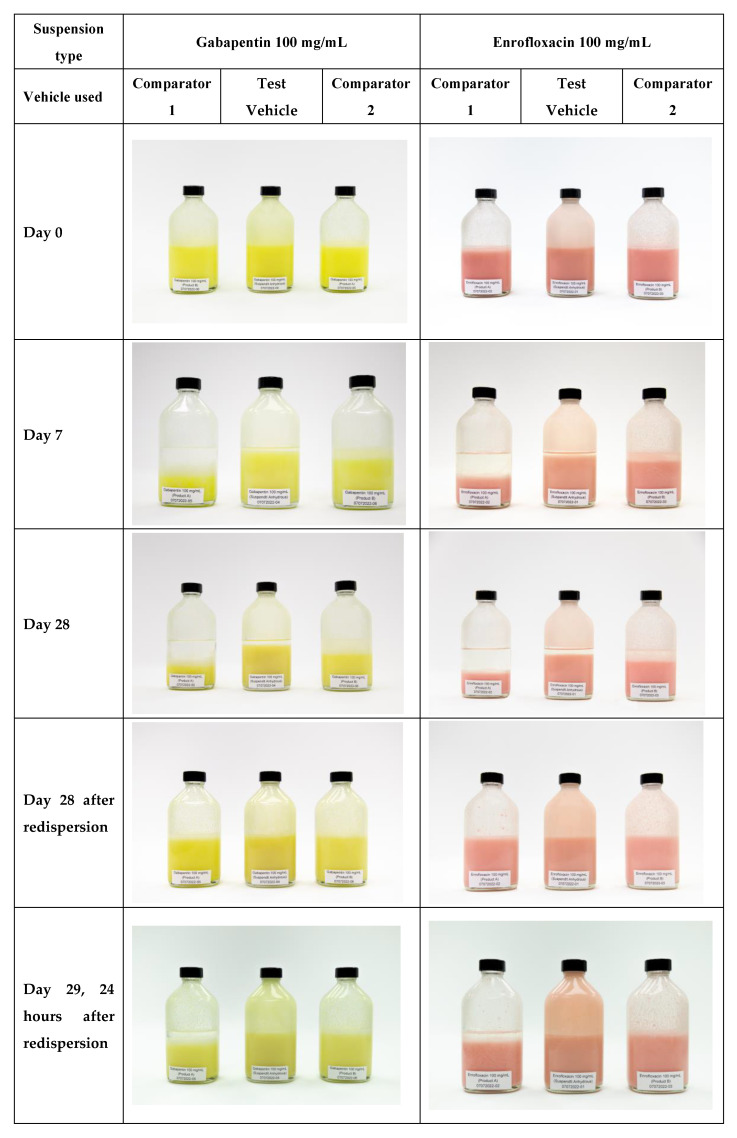
Sedimentation of gabapentin and enrofloxacin anhydrous suspensions over 28 days and after redispersion (gabapentin 100 mg/mL anhydrous suspension (yellow color), enrofloxacin 100 mg/mL anhydrous suspension (pink color)).

**Figure 2 pharmaceutics-15-02642-f002:**
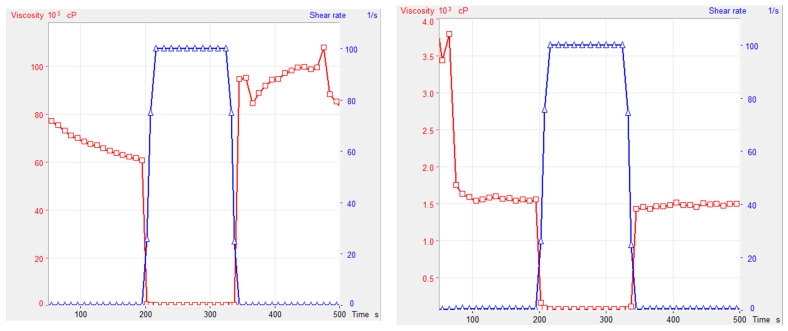
Thixotropic properties of Test Vehicle (**left**) and Water-Based Test Vehicle (**right**) at room temperature. (Red lines represent the change in viscosity and blue lines represent the change in shear rate.).

**Figure 3 pharmaceutics-15-02642-f003:**
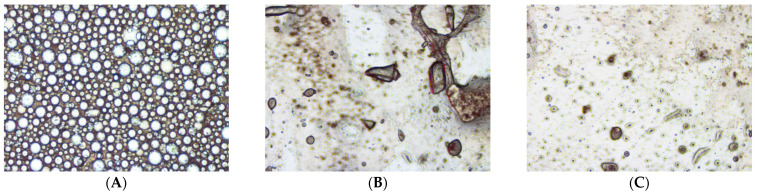
Microscopic evaluation of droplet size formation (4× objective lens) for metronidazole 50 mg/mL anhydrous suspension mixed with water at 1:1 ratio (*v*/*v*) (**A**—Test Vehicle, **B**—Comparator 1, **C**—Comparator 2).

**Figure 4 pharmaceutics-15-02642-f004:**
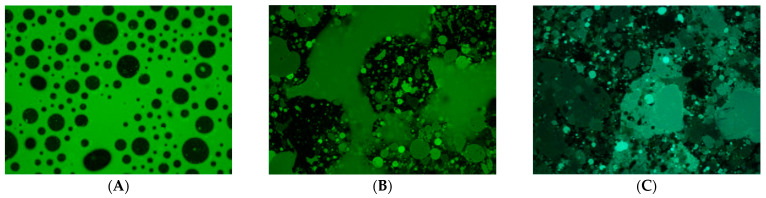
Fluorescence microscopy (fluorescent light) at 4× objective lens for fluorescein sodium 1% (**A**—Test Vehicle, **B**—Comparator 1, **C**—Comparator 2).

**Figure 5 pharmaceutics-15-02642-f005:**
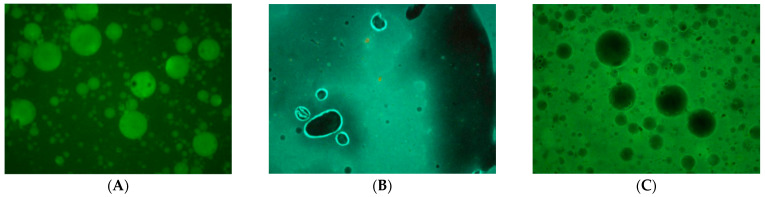
Fluorescence microscopy (fluorescent light) at 10× objective lens for curcumin 1% (**A**—Test Vehicle, **B**—Comparator 1, **C**—Comparator 2).

**Figure 6 pharmaceutics-15-02642-f006:**
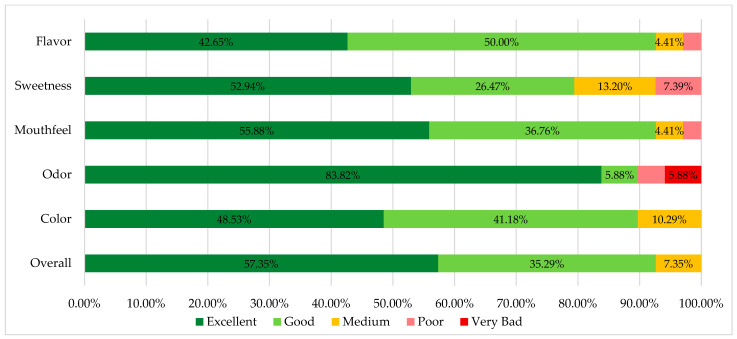
Cumulative chart of participants’ ratings of the organoleptic characteristics of the anhydrous Test Vehicle.

**Table 1 pharmaceutics-15-02642-t001:** Mean potency of compounded oral suspensions prepared using different methods.

Oral Suspensions	Method of Preparation	Mean Potency, %	RSD, %	AV, % ^1^
Chloroquine phosphate 100 mg/5 mL	Mortar and pestle	104.00	1.05	5.03
Doxycycline 50 mg/mL	Mortar and pestle	101.00	1.32	5.68
Doxycycline 100 mg/mL	Mortar and pestle	100.00	1.50	5.10
Doxycycline 100 mg/mL	FlackTek SpeedMixer	102.00	1.89	5.03
Enrofloxacin 10 mg/mL	Mortar and pestle	99.00	0.81	2.45
Enrofloxacin 100 mg/mL	Mortar and pestle	100.00	0.57	2.87
Ketotifen 1 mg/mL	Mortar and pestle	101.10	1.00	5.00
Metronidazole 50 mg/mL	Mortar and pestle	104.60	1.38	6.41
Metronidazole 50 mg/mL	Mortar and pestle	104.20	1.50	6.29
Nifedipine 4 mg/mL	Mortar and pestle	96.60	0.69	3.55
Phenoxybenzamine HCl 2 mg/mL	Mortar and pestle	95.50	1.60	6.83
Tretinoin 10 mg/mL	Mortar and pestle	100.74	0.88	4.34
Tretinoin 10 mg/mL	FlackTek SpeedMixer	102.00	1.20	3.37

^1^ Maximum allowed acceptance value L_1_ = 15.0%.

**Table 2 pharmaceutics-15-02642-t002:** Droplet size diameter distribution by selected percentiles for ibuprofen 40 mg/mL in two compounded anhydrous oral suspensions versus the commercial product of reference.

Suspensions	Droplet Size Diameter (μm)			
	90%	50%	10%	Mean ± Std Dev
Ibuprofen 40 mg/mL (Test Vehicle)	65.422	48.519	8.097	43.162 ± 0.898
Ibuprofen 40 mg/mL (reference commercial product)	73.228	36.356	9.891	38.652 ± 0.142
Ibuprofen 40 mg/mL (Comparator 2)	255.273	192.657	96.671	183 ± 0.034

## Data Availability

Data are contained within the article and Appendix A.

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
