# Peer review of "Analysis of the Physical Characteristics of an Anhydrous Vehicle for Compounded Pediatric Oral Liquids"

_pharmaceutics, 2023, doi:10.3390/pharmaceutics15112642_

Round 1

Reviewer 1 Report

Comments and Suggestions for Authors

Line 49: "omeprazole, )," is it a typo?

Line 47: What is "most common" based on? Is it children? Adults with dysphagia?

I think omeprazole would be esomeprazole for oral use.

Line 79: How about PVC or silicon tube?

Lines 100-118: How about a table to make it easier to understand each experimental method and its purpose?

Line 229: Did you obtain written consent from the test subjects for the preference test? Did you apply to an ethics committee?

Also, you are conducting a preference test for the test vehicle only, but wouldn't it be meaningful if you don't evaluate the preference when the actual drug (API) is mixed with the test vehicle?

Results: what would be the main factor affecting the sedimentation rate (would it be excipients?). If the main factors are determined, would it be possible to categorize compounds that are more likely to sediment and compounds that are less likely to sediment as model compounds?

For each of the properties tests, the test compounds are different, what were the criteria used to select them?

Discussion

Lines 385-388: Is there any relationship between similar dispersion in gastric fluid and equivalent bioavailability? I don't think this study will provide any information on the equivalence of bioavailability.

Conclusion

Line 406-.

As the author says, you certainly need to select an appropriate dissolution solution, but it is not at all clear in this study what to select.

For example, is it possible to summarize the paper in terms of setting up a basic test for selecting appropriate vehicles, or how to categorize them by physical properties so that appropriate vehicle can be selected?

I think this paper only provides a narrative of a real-world example.

Reviewer 2 Report

Comments and Suggestions for Authors

The paper is well-written and has a few technical errors. Unfortunately, the authors did not highlight the scientific significance. The work seems more like a professional than a scientific work, especially for such an exceptional magazine.

It deals with current issues, but the scientific contribution is not highlighted.

1. Regarding a large number of keywords, remove those that are repeated in the abstract and title, see the instructions

2. In the abstract, you measured the size of the particles, and in the keywords, the size of the drops. If you measured both it is fine, if not you will have to correct it. The size of the particles in dispersed systems is what suspensions are, more precisely when we have a solid phase dispersed in a liquid in which it does not dissolve. Drop size in emulsions, more precisely when one liquid phase is dispersed in another, in which it does not mix.

3. Consider whether the term sedimentation is correct when talking about emulsion instability or phase separation. Sedimentation is characteristic of suspensions.

4. Did you do a stability study during the application? If you opened bottles, simulated applications...

5. Row 202, how is the size distribution of emulsions? Distribution of most drops of the internal phase of emulsions, formed after emulsification.

6. 2.4. And 2.5. very fleshy emulsions, suspensions, drop size, particles, think about it. Proofread throughout the paper.

7. Did you agree with the ethics commission for the taste evaluation?

8.  There is no explicit comparison and referencing with other works during the discussion.

Comments on the Quality of English Language

 Minor editing of the English language required

Reviewer 3 Report

Comments and Suggestions for Authors

The manuscript “Analysis of the Physical Characteristics of an Anhydrous Vehicle for Compounded Pediatric Oral Liquids” concerns Suspendit Anhydrous, a new, proprietary anhydrous vehicle for the preparation of API suspensions. Interestingly, the drug-vehicle systems are self-emulsifying, giving rise to fine droplets when in contact with body fluids, thus positively contributing to the API’s bioavailability. Many suspensions of drugs belonging to various pharmacological groups were prepared and extensively characterized. Indeed, they were tested under the respects of sedimentation, viscosity and thixotropy, concentration uniformity, droplets size of the emulsion formed in simulated gastric fluid, palatability. The performances of Suspendit Anhydrous are compared with those of two proprietary anhydrous vehicles and to a commercial ibuprofen preparation, regarding droplet size of the emulsion obtained in the simulated gastric fluid. The outcomes of the investigations look very promising for the practical applications of the new vehicle.

As a reader, I find lack of information about the composition of the tested vehicles, even for the commercial ibuprofen preparation. I suggest to add as much information as possible about it, of course, compatibly with industrial property issues.

A far as SuspendIt is concerned, in the paper Physicochemical Stability of Extemporaneously Prepared Oral Suspension of Fluconazole 50 mg/mL in SuspendIt™ K. Ip, A. Shan, M. Carvalho, S. Baker and D. Banov Pharm Technol Hosp Pharm 2018; 3(2): 101–112 it is reported the following composition “SuspendIt™ is a proprietary oral suspending vehicle indicated in paediatrics that includes the following ingredients: water, amorphophallus konjac root powder, monk fruit extract (natural sweetener), xanthan gum, potassium sorbate, sodium benzoate, citric acid and disodium EDTA.”

On the PCCA website https://www.pccarx.com/products/PCCASUSPENDIT%C2%AEANHYDROUS/30-5176/PROPRIETARYBASES

it is reported:

For SuspendIt Anhydrous Oral Base Only

Protein 0.3%

Fat 87.7%

Carbohydrates 7.7%

Calories 821 Cal/100 gm

 I suggest the authors to slightly modify both the Abstract and the Introduction sections, soon stating that they are testing “the PCCA's new, patent-pending oral suspending base, SuspendIt Anhydrous, which is ideal for active pharmaceutical ingredients (APIs) that are unstable in water or have incompatibilities with existing aqueous vehicles.” https://finance.yahoo.com/news/pcca-introduces-innovative-compounding-suspendit-192800470.html?guccounter=1&guce_referrer=aHR0cHM6Ly93d3cuZ29vZ2xlLmNvbS8&guce_referrer_sig=AQAAADc9V5H38F7o-5wUolbdQpN2O-mCKJd5TGTFgli7doaMY1Nybk-X9WcpCbLAE21-SEXTDQBYPcDMLupozzE6pEr5xj9Vr2StweZDJqjxDND-IwAFO7Jm7jkYJNwNVuVCM9QQAD4JIV8C6F6NtJZwNaT5JHnOurHO5glsg7dIkoTD

After all, the tradename “Suspendit” is present even in the titles of refs. 45 and 46

Furthermore, at line 63 I would find useful to add that important anhydrous oral vehicles are non aqueous emulsions, ref. 28 and references therein.

There a few minor issues:

line 58: better “molecules” than “chemical entities”, alternatively “mixtures”, if more appropriate

line 120: add “Suspendit Anhydrous” after “new developed anhydrous oral vehicle”

line 176: add the rheometer characteristics

line 225: add the information of the light source employed for fluorescence excitation

lines 360-363: it is not clear whether the sentence is referred to Comparator 2 or to SuspendIt Anhydrous, also because the, previously mentioned, lack of information about the vehicles’ composition.

Supplementary Materials:

at the link https://player.vimeo.com/video/76205410 I got a nice video on windsurfing!

Round 2

Reviewer 1 Report

Comments and Suggestions for Authors

No additional comments

Reviewer 2 Report

Comments and Suggestions for Authors

Accept in present form

Reviewer 3 Report

Comments and Suggestions for Authors

The detailed information added to the revised version of the manuscript contributes to a better understanding of the performances and behavior of the anhydrous vehicle, subject of the study, and to the replicability of the experiments.